# Dual Inhibition of EGFR and IGF-1R Signaling Leads to Enhanced Antitumor Efficacy against Esophageal Squamous Cancer

**DOI:** 10.3390/ijms231810382

**Published:** 2022-09-08

**Authors:** Jia Kang, Zanzan Guo, Haoqi Zhang, Rongqi Guo, Xiaofei Zhu, Xiaofang Guo

**Affiliations:** 1Department of Pathogenic Biology, School of Basic Medical Sciences, Xinxiang Medical University, Xinxiang 453003, China; 2Xinxiang Key Laboratory of Pathogenic Biology, Xinxiang 453003, China; 3School of Laboratory Medicine, Xinxiang Medical University, Xinxiang 453003, China; 4Xinxiang Key Laboratory of Tumor Microenvironment and Immunotherapy, Xinxiang 453003, China; 5Henan Key Laboratory of Immunology and Targeted Drugs, Xinxiang 453003, China; 6Xinxiang Molecular and Immunodiagnostics Research Center for Engineering Technology, Henan Collaborative Innovation Center of Molecular Diagnosis and Laboratory Medicine, Xinxiang 453003, China

**Keywords:** dual inhibition, esophageal cancer, gefitinib, lapatinib, linsitinib

## Abstract

Both the epidermal growth factor receptor (EGFR) and insulin-like growth factor 1 receptor (IGF-1R) have been implicated in the development of cancers, and the increased expression of both receptors has been observed in esophageal cancer. However, the tyrosine kinase inhibitors of both receptors have thus far failed to provide clinical benefits for esophageal cancer patients. Studies have confirmed the complicated crosstalks that exist between the EGFR and IGF-1R pathways. The EGFR and IGF-1R signals act as mutual compensation pathways, thereby conveying resistance to EGFR or IGF-1R inhibitors when used alone. This study evaluated the antitumor efficacy of the EGFR/HER2 inhibitors, gefitinib and lapatinib, in combination with the IGF-1R inhibitor, linsitinib, on the esophageal squamous cell carcinoma (ESCC). Gefitinib or lapatinib, in combination with linsitinib, synergistically inhibited the proliferation, migration, and invasion of ESCC cells, caused significant cell cycle arrest, and induced marked cell apoptosis. Their combination demonstrated stronger inhibition on the activation of EGFR, HER2, and IGF-1R as well as the downstream signaling molecules. In vivo, the addition of linsitinib to gefitinib or lapatinib also potentiated the inhibition effects on the growth of xenografts. Our results suggest the next clinical exploration of the combination of gefitinib or lapatinib with linsitinib in the treatment of ESCC patients.

## 1. Introduction

According to the data from GLOBOCAN 2020, esophageal cancer is the eighth most common type of cancer and constitutes the sixth leading cause of cancer deaths worldwide [1]. Esophageal squamous cell carcinoma (ESCC) is the most prevalent histological type in Eastern Europe and Asia, while North America and Western Europe have predominantly esophageal adenocarcinoma (EAC) [2]. China has the highest incidence of esophageal cancer in the world, and more than 90% of the patients are ESCC [3]. Esophageal cancer is highly aggressive and is rarely found before it has advanced or metastasized. The prognosis of patients with metastatic esophageal cancer is extremely poor, with a median overall survival of 4–6 months [4]. Although the combined modality therapy and systemic therapy for esophageal cancer have made great progress in recent years, the 5-year survival rate of patients (about 15–25%) has not significantly improved [5]. Targeted therapy has become an important treatment strategy for many malignant tumors; however, few targeted drugs have been approved by the FDA for treating esophageal cancer. At present, only ramucirumab (VEGFR-targeting) has been approved as a second-line treatment for EAC, and trastuzumab deruxtecan (HER2-targeting) has been approved as a first-line treatment in combination with chemotherapy in HER-2-positive gastric cancer and gastroesophageal adenocarcinoma patients [6].

Both the insulin-like growth factor 1 receptor (IGF-1R) and epidermal growth factor receptor (EGFR) were implicated in the development, progression, metastasis, and chemotherapy resistance of a variety of cancers [7,8]. The IGF-1R signaling system comprises two ligands (IGF-1 and IGF-2), two receptors (IGF-1R and IGF-2R), and six circulating IGF-binding proteins (IGFBPs 1–6) [9]. IGF-1R is a glycoprotein composed of two extracellular α subunits that bind IGF-1 preferentially and with a lesser affinity to IGF-2 and insulin, and two β subunits contain the tyrosine kinase domain responsible for the activation of the two main downstream signaling pathways (the PI3K/AKT and Ras/MAPK pathways) that promote cell growth, transformation, migration, and survival [9]. In recent decades, a large body of evidence has supported the key role of IGF-1R signaling in the transformation of cells, cancer cell proliferation, and cancer metastasis. The overexpression of IGF-1 and IGF-1R has been observed in many tumor cell lines and tissues, and their overexpression is closely related to the poor prognosis of patients [10]. Our previous studies found that IGF-1R was also highly expressed in esophageal cancer, and the bispecific fusion protein targeting EGFR and IGF-1R that we constructed had very significant inhibitory activity against esophageal cancer in vitro and in vivo [11]. Therapeutic strategies targeting IGF-1R, including the use of monoclonal antibodies (figitumumab, ganitumumab, dalotuzumab, etc.), tyrosine kinase inhibitors (TKIs) (linsitinib (OSI-906), NVP-ADW742, BMS-754807, etc.), and IGF ligand neutralizing antibodies (MEDI-573 and BI 836845), have been explored in preclinical studies and clinical trials. Despite the promising preclinical reports, clinical trials did not provide meaningful benefits with IGF-1R inhibitors [12,13,14,15,16,17,18,19], and none of them have been approved for clinical use.

The epidermal growth factor receptor (EGFR) and human epidermal growth factor receptor 2 (HER2) are the members of the ErbB family of receptor tyrosine kinases. The ErbB family also plays important roles in cell proliferation, differentiation, and survival, and their overexpression and mutation were observed in a majority cancers, including esophageal cancer, which made them vital targets for treating esophageal cancer [6,7]. Therapies targeting the EGFR family, including monoclonal antibodies, antibody-drug conjugates (ADCs), and TKIs, have been evaluated in many clinical trials for esophageal cancer patients. Among them, only trastuzumab deruxtecan (Enhertu), an HER2-targeting ADC, led to significant improvements in response and overall survival compared with standard therapies for patients with HER2-positive gastric and gastroesophageal adenocarcinoma [20]. The monoclonal antibodies (e.g., cetuximab) and TKIs (e.g., gefitinib and lapatinib) did not show superior activity to chemotherapy alone [21,22,23,24].

The resistance to monotherapies targeting IGF-1R and EGFR/HER2 remains a major challenge. The disappointing efficacy of these inhibitors may be due to the lack of validated predictive biomarkers for patient selection and to the existence of compensatory signaling pathways. Studies have shown that there are complicated crosstalks between the EGFR and IGF-1R signal pathways. Firstly, the two receptors can directly form EGFR/IGF-1R or ErbB2/IGF-1R heterodimers, and the cell surface interaction can also be indirectly mediated by the activity of G protein coupled receptors (GPCRs) [25]. Secondly, they share the same downstream signaling components that regulate the expression of ligands, receptors, and IGFBPs. IGF-1R signaling may increase the autocrine production of EGFR ligands, while EGFR signaling, in turn, may regulate the availability of IGF-1 through its effects on IGFBPs [25]. The crosstalk between EGFR/HER2 and IGF-1R contributes to the resistance to EGFR and IGF-1R monotherapies [26]. Based on this rationale, we hypothesized that the dual inhibition of EGFR/HER2 and IGF-1R will achieve superior activity to targeting either receptor alone.

In this study, the EGFR inhibitor, gefitinib, and the EGFR and HER2 dual inhibitor, lapatinib, combined with the IGF-1R inhibitor, linsitinib, were performed to treat four ESCC cells, and their effects on cell viability, cell cycle distribution, cell apoptosis, cell invasion, and migration as well as the in vivo efficacy in the transplanted mouse model were measured. The aim of the study was to provide preclinical data and a basis for the treatment of esophageal cancer with lapatinib or gefitinib combined with linsitinib and to explore a new scheme for the targeted therapy of esophageal cancer.

## 2. Results

### 2.1. Lapatinib or Gefitinib in Combination with Linsitinib Is Synergistic in the Growth Inhibition of ESCC Cells

As shown in Figure 1b, lapatinib, gefitinib, and linsitinib alone significantly inhibited the proliferation of four ESCC cells. Their IC50 values against four different ESCC cells expressing variable levels of EGFR, HER2, and IGF-1R were quite similar, except for the KYSE150 and KYSE510 cells. The two cells are relatively insensitive to linsitinib (Figure 1a,b, Table 1).

To detect the interaction between the EGFR/HER2 inhibitor and the IGF-1R inhibitor, lapatinib or gefitinib was mixed with linsitinib at a fixed ratio (1:4) to treat the ESCC cells for 48 h. The combination of lapatinib or gefitinib and linsitinib displayed a potentiated inhibition of proliferation compared to either single agent (Figure 1c). The combination index (CI) values between lapatinib/gefitinib and linsitinib at different dose-effect levels—ED50 (effective dose of 50% response), ED75 (that of 75% response), and ED90 (that of 90% response)—were calculated by using the CompuSyn software (version 1.0) (Paramus, NJ, USA) which was created by Ting-Chao Chou and Nick Martin and are shown in Table 2. The CI values were all less than 1, which indicates that lapatinib or gefitinib in combination with linsitinib was generally synergistic in four ESCC cell lines (Figure 1d).

### 2.2. Lapatinib or Gefitinib in Combination with Linsitinib Induces Enhanced G1 Arrest and Cell Apoptosis in ESCC Cells

Our previous study found that lapatinib arrested ESCC cells in the G1 phase; therefore, the cell cycle distribution was analyzed after treatment with lapatinib (2.5 μmol/L), gefitinib (2.5 μmol/L), and linsitinib (10 μmol/L) alone or the combinations. The flow cytometry data revealed that, in KYSE150 and KYSE450 cells, EGFR and/or HER2 inhibitors (gefitinib and lapatinib) treatment resulted in significant G1 arrest (*p* < 0.001 vs. control). Although the cells in the G1 phase were also increased after exposure to linsitinib, the effect of G1 arrest was far less than that of lapatinib and gefitinib (*p* < 0.05 vs. lapatinib, *p* < 0.001 vs. gefitinib). The combinations of lapatinib/gefitinib and linsitinib also caused G1 arrest (*p* < 0.001 vs. control); however, it was not as significant as that of lapatinib or gefitinib alone, but it was more significant than that of linsitinib alone. In TE-7 cells, only gefitinib, not lapatinib or linsitinib, was able to cause G1 arrest (*p* < 0.001 vs. control), and lapa+lins and gefi+lins had the same effect (*p* < 0.01 lapa+lins vs. control, *p* < 0.001 gefi+lins vs. control). The G1 arrest in the lapa+lins treatment group increased compared to the lapatinib-alone group (*p* < 0.05), while the G1 arrest effect of gefi+lins did not increase correspondingly compared with that of gefitinib alone. However, the G1 block in the two drug combination groups increased significantly compared with that of linsitinib alone (*p* < 0.001 vs. lapa+lins and gefi+lins). KYSE510 cells are very special; only gefitinib treatment can cause G1 arrest, while lapatinib alone, linsitinib alone, lapa+lins, and gefi+lins cannot. Despite the statistical differences, the proportions of cells in the G1 phase after treatment by lapatinib, linsitinib, lapa+lins, or gefi+lins were smaller than those of the control group (Figure 2a and Appendix A).

Inducing apoptosis is an important mechanism for TKIs to exert their antitumor effect; thereafter, we evaluated the efficacy of three single agents alone or in combination on cell apoptosis. Figure 2b shows that lapatinib and gefitinib significantly and dose-dependently induced ESCC cell apoptosis. When combined with linsitinib, further enhanced apoptosis was observed in four ESCC cells (Figure 2b and Appendix A). Caspase-3 is a key executor of cell apoptosis because it partially or completely cleaves many key proteins. Thereafter, we detected the expression of cleaved caspase-3 after treatment by lapatinib, gefitinib, and linsitinib alone or in combination in four ESCC cells by western blot. Figure 3c revealed that single lapatinib, gefitinib, and linsitinib treatment resulted in the increase in cleaved caspase-3 in four ESCC cells, and its expression was further increased after exposure to the combination of lapatinib/gefitinib and linsitinib. This result confirmed that lapatinib or gefitinib in combination with linsitinib caused enhanced proapoptotic activity against ESCC cells (Figure 3c).

### 2.3. Lapatinib or Gefitinib in Combination with Linsitinib More Potently Inhibited the Invasion and Migration of ESCC Cells

Invasion and migration are the important characteristics of malignant tumors. Next, we examined the effects of the EGFR/HER2 inhibitor and IGF-1R inhibitor alone or in combination on cell invasion and migration. A wound healing assay and transwell assay were used to evaluate the cell migration, and a transwell chamber coated with matrigel was used to evaluate the cell invasion. As shown in Figure 3, the area of scratch in the lapatinib, gefitinib, and linsitinib treatment group was significantly smaller than that of the control group at 24 h and 48 h for KYSE150 and TE-7 cells (*p* < 0.05). The combination of lapatinib or gefitinib with linsitinib further inhibited the healing of the scratches (*p* < 0.05 vs. single drug treatment groups).

The transwell assay showed similar results to the wound healing assay. The number of KYSE150 and TE-7 cells migrating and invading from the upper chamber to the lower chamber decreased significantly after treatment with lapatinib, gefitinib, and linsitinib alone (*p* < 0.01 vs. control), and the cell number further decreased in the lapatinib plus linsitinib and gefitinib plus linsitinib treatment groups, which was significantly different from that in the single drugs treatment group (*p* < 0.001, Figure 4a–f).

### 2.4. Lapatinib or Gefitinib in Combination with Linsitinib Synergistically Inhibited the EGFR/HER2 and IGF-1R Signaling Activation

From the results of western blot assays, we found that lapatinib or gefitinib (EGFR and/or HER2 inhibitors) treatment alone could decrease the phosphorylation of EGFR and HER2 (lane 5 and lane 6) that was stimulated by EGF and IGF-1, and linsitinib (IGF-1R inhibitor) treatment alone could decrease the phosphorylation of IGF-1R (lane 7) in four ESCC cells. However, the linsitinib treatment showed no inhibitory effects on the phospho-EGFR and phospho-HER2 level (KYSE150 and TE-7 cells) or even increased the expression of phospho-EGFR and phospho-HER2 (KYSE450 cells). Only KYSE510 cells were an exception. Correspondingly, lapatinib or gefitinib treatment increased the phospho-IGF-1R level (KYSE150, KYSE450, and TE-7 cells), but in KYSE510 cells, gefitinib treatment inhibited the phosphorylation of IGF-1R. For the two main downstream molecules, AKT and p44/42MAPK (ERK), gefitinib alone, but not lapatinib and linsitinib, inhibited their activation in KYSE150, KYSE450, and KYSE510 cells. In TE-7 cells, the three drugs alone could not reduce the expression of phospho-AKT and phospho-ERK, nor could the combination of lapatinib and linsitinib (Figure 5).

In the two drugs combination therapy, in general, the combination of gefitinib and linsitinib exerted a stronger inhibitory effect on signal transduction. In the four ESCC cells, the combination of gefitinib and linsitinib could not only significantly reduce the activation of EGFR, HER2, IGF-1R, AKT, and ERK, but it also had a stronger inhibitory effect than any single drugs. However, the combination of lapatinib and linsitinib has less of an inhibitory effect on the EGFR/HER2 and IGF-1R signal pathways than that of gefitinib plus linsitinib. Moreover, in TE-7 cells, it did not inhibit the phosphorylation of AKT, and it significantly increased the phospho-ERK level (Figure 5).

### 2.5. Lapatinib or Gefitinib Combined with Linsitinib More Potently Inhibited the Tumor Growth In Vivo

The in vivo efficacy of lapatinib, gefitinib, and linsitinib alone or in combination was evaluated in the KYSE450 xenograft model in nude mice. Lapatinib (100 mg/kg), gefitinib (50 mg/kg), and linsitinib (30 mg/kg) were given to tumor-bearing mice for 18 days (from day 12 to day 30) through oral gavage. Lapatinib alone yielded minimum growth inhibition, with a tumor growth inhibition rate (TGI) of 37.4% (*p* < 0.001 vs. control). The in vivo inhibitory activity of linsitinib alone was moderate, with a TGI of 54.2% (*p* < 0.001 vs. control, Figure 6a,c). Lapatinib in combination with linsitinib markedly inhibited the growth of xenografts (TGI of 66.8%), which was more effective than lapatinib, but was not significant different from linsitinib (*p* < 0.001 vs. lapatinib; *p* = 0.08 vs. linsitinib). To our surprise, gefitinib alone at 50 mg/kg demonstrated a very potent tumor growth inhibiting efficacy (TGI of 85.7%), which was not only stronger than that of lapatinib and linsitinib (*p* < 0.001 vs. lapatinib or linsitinib) but also stronger than that of lapatinib plus linsitinib (*p* < 0.001). Furthermore, gefitinib combined with linsitinib yielded the strongest in vivo growth inhibition (TGI of 93.7%) (*p* < 0.001 vs. lapatinib, linsitinib, and lapa+lins), but there was no significant difference between the single gefitinib group and the gefitinib+linsitinib group (*p* > 0.99).

During the experiment, the changes in the body weight of mice were observed to evaluate the toxicity of the drugs. Figure 6b revealed that lapatinib and gefitinib alone had barely no effect on the mice weight. The mice weight in the linsitinib group and the combined treatment groups decreased briefly at the initial stage of administration but gradually recovered at the later stage. At the end of the experiment, there were no differences compared with the control group and the single drug treatment groups. There was no mice death, nor were any other toxic signs observed during the entire experiment, which indicated the good tolerance of the mice to the treatments.

To clarify the effects on signal transduction in vivo, the xenograft tumor tissues taken from the mice on day 33 were homogenized and lysed, and the total proteins were extracted to perform the western blot analysis. The results showed that treatment with gefitinib and gefi+lins decreased the phospho-EGFR significantly, whereas no changes in the lapatinib, linsitinib, and lapa+lins treatment groups were observed. The phosphorylation of HER2 was only inhibited strongly by gefi+lins exposure; lapatinib, gefitinib, linsitinib, and lap+lins did not show a marked inhibitory effect. Both lapatinib and gefitinib did not play roles in the activation of IGF-1R, while the addition of linsitinib blocked the phosphorylation of IGF-1R. In the inhibition of the phosphorylation of AKT and ERK, gefitinib and gefi+lins were also stronger than lapatinib, linsitinib, and lapa+lins. With the exception of IGF-1R, lapatinib, gefitinib, and linsitinib alone or in combination had little effect on the total protein levels of the main signal molecules. The total IGF-1R level decreased significantly in the gefitinib-alone and the combination (lapa+lins and gefi+lins) treatment groups (Figure 6d).

## 3. Discussion

Surgery, radiotherapy, and chemotherapy are still the most common methods for treating esophageal cancer. For localized esophageal cancer, endoscopic resection and combined modality therapy including preoperative chemoradiation or perioperative chemotherapy are common options, while for advanced or metastasis cases, systemic therapy with the cisplatin and fluoropyrimidines (5-fluorouracil or capecitabine) regimen is the standard treatment for palliation [4,5]. These traditional strategies improve the survival and outcome, but they are totally insufficient and unsatisfactory for the treatment of esophageal cancer. Therefore, it is essential to investigate new treatment methods, such as targeted therapy and immunotherapy. Immunotherapy—mainly, checkpoint inhibitors—has shown efficacy in treating advanced esophageal cancer [4]. FDA-approved nivolumab plus chemotherapy and nivolumab plus the monoclonal antibody ipilimumab are the first-line treatments for adults with previously untreated, unresectable advanced, recurrent, or metastatic ESCC based on the data from the CheckMate-648 phase III clinical trial [27]. Another PD-1 inhibitor, pembrolizumab, in combination with fluoropyrimidine plus platinum-based chemotherapy was also approved for patients with metastatic or locally advanced esophageal cancer or gastroesophageal cancer who are not suitable for surgical resection or definitive radiotherapy based on the KEYNOTE-590 clinical trial [28].

Compared to immunotherapy, less progress was made in the targeted therapeutic drugs for esophageal cancer, and very few drugs were approved. The limited efficacy of EGFR and HER2 inhibitors due to the primary and acquired resistance and the failure of IGF-1R inhibitors in clinical trials were largely attributed to the complexity of receptor tyrosine kinase (RTK) pathway signaling, including compensatory pathway activation through other RTKs [26,29,30]. As we discussed before, there are complicated crosstalks between the EGFR/HER2 and IGF-1R pathways through various mechanisms. Interactions between IGF-1R and EGFR signaling have been proven to contribute to the development of resistance to anti-EGFR therapies. Moreover, the dual inhibition of EGFR/HER2 and IGF-1R signaling did produce stronger antitumor effects than either monotherapies in pancreatic cancer, ovarian cancer, colon cancer, etc. [30,31,32,33]. Therefore, we examined the antitumor activity of the EGFR/HER2 inhibitor (gefitinib and lapatinib) combined with the IGF-1R inhibitor (linsitinib) on the ESCC.

Gefitinib, lapatinib, and linsitinib showed similar inhibitory activity on the proliferation of four ESCC cells, although the four ESCC cells had variable levels of EGFR, HER2, and IGF-1R, as revealed by MTT assays. The IC50 values of linsitinib against the KYSE150 and KYSE510 cells were 22.42 μmol/L and 15.83 μmol/L, respectively, and the other IC50 values ranged from 4.605 μmol/L to 8.489 μmol/L. The results indicated that there was no correlation between drug sensitivity and the expression levels of EGFR, HER2, and IGF-1R. When the two drugs are in combination, drug concentrations should be optimized to achieve the synergistic effect. In the present study, five concentrations across their IC50 values were selected, and lapatinib or gefitinib was mixed with linsitinib at a 1:4 ratio. At this ratio, the CI values of lapatinib or gefitinib in combination with linsitinib at ED50, ED75, and ED90 were all less than 1, indicating the significant synergistic effect between the two drugs. However, at other proportions (such as 1:2), the combination of the two drugs showed additive or antagonistic effects (Appendix A). Therefore, it will not always show a synergistic effect at any concentration and proportion. When drugs are used together, their concentration and proportion must be thoroughly optimized.

Lapatinib and gefitinib showed significant G1 arrest in the KYSE150 and KYSE450 cells in the cell cycle assay. Linsitinib alone also caused G1 arrest, although its effect was not as significant as that of lapatinib and gefitinib. However, when they were used together, there was no stronger G1 arrest observed. Moreover, the G1 arrest in the combination groups (lapa+lins and gefi+lins) was weaker than that in the lapatinib-alone and gefitinib-alone groups. This indicated that there were no synergistic or additive effects on cell cycle arrest. We thought that this may be caused by the high concentration of lapatinb and gefitinib. The cell cycle arrest caused by them at 2.5 μmol/L was already very significant, so, when they were combined with linsitinib, the effect on cell cycle arrest could not be further improved. Synergistic effects on cell cycle arrest may occur when lapatinib and gefitinib are at lower concentrations.

When detecting the effect of drugs on EGFR/HER2 and IGF-1R signaling, different results were observed in different drug treatment groups and different cell lines. For example, the activation of EGFR/HER2 and IGF-1R was inhibited by their corresponding inhibitors (gefitinib/lapatinib and linsitinib) in KYSE510 cells, but in KYSE450 cells, linsitinib significantly increased the phosphorylated EGFR and HER2, and gefitinib and lapatinib also significantly increased the phosphorylation of IGF-1R. This is possibly because the compensatory IGF-1R pathway was activated for signal transduction after the EGFR/HER2 signaling pathway was inhibited, which is one of the reasons for the poor efficacy of RTKs inhibitors used alone. When lapatinib/gefitinib and linsitinib were used in combination, the activation of EGFR/HER2 or IGF-1R was reversed.

Gefitinib is an inhibitor of EGFR, while lapatinib is a dual inhibitor of EGFR and HER2, but gefitinib has a stronger inhibitory effect on the EGFR/HER2 signaling pathway than lapatinib. Moreover, the in vivo efficacy of gefitinib against KYSE450 xenografts was also more potent than that of lapatinib, linsitinib, or even the lapatinib plus linsitinib treatment. To try to understand the mechanisms, we detected the expression of phospho-EGFR, -HER2, -IGF-1R, -AKT, and -ERK as well as their total protein levels in the xenograft tumor tissues by western blot analysis. The results revealed that the gefitinib and gefi+lins treatments resulted in a marked reduction in phospho-EGFR, whereas no changes in the lapatinib, linsitinib, and lapa+lins treatment groups were observed. In the inhibition of phospho-AKT and phospho-ERK, gefitinib and gefi+lins were also stronger than lapatinib, linsitinib, and lapa+lins. These results demonstrated that gefitinib inhibits the signaling pathway more effectively in vivo, and this may be one of the reasons why gefitinib and gefi+lins showed superior antitumor efficacy. In addition, we sequenced the mutation prone regions of EGFR (exon 19–21) and HER2 (exon 20, 21). However, no mutations, insertions, or deletions were found in these regions (Appendix A). Kwak et al. believed that EGFR and KRAS rarely mutated in esophageal cancer, which is one of the reasons why few patients benefit from anti-EGFR therapy [34]. The results of this study revealed that the efficacy of gefitinib in ESCC is not only determined by the mutations of EGFR and KRAS. Therefore, the screening of prediction molecules is the next focus of our group, in order to more accurately find ESCC patients who can benefit from gefitinib treatment. Whether gefitinib had the same strong inhibitory effects on other ESCC cells is another focus of further studies. On the other hand, we believe that the pharmacokinetic characteristics of lapatinib and gefitinib also affect their activity. According to the literature, the absolute bioavailability is ~60% for gefitinib, and food has only a clinically non-significant effect on gefitinib exposure. For lapatinib, its bioavailability is low since food has an extraordinary effect on bioavailability. The elimination half-lives for gefitinib and lapatinib were 48 h and 24 h, respectively, which indicated that gefitinib stays in the body and exerts its effect for a longer time [35]. Moreover, gefitinib preferably distributed into tumor tissues after it was administrated. The tumor cell/plasma ratio was 11-fold, as was the skin/plasma ratio in mice bearing human tumor xenografts [36]. Therefore, the better pharmacokinetic characteristics of gefitinib may also be related with its better in vivo activity.

In addition to EGFR and HER2, IGF-1R interacts with ErbB3, another member of the ErbB family. A study from Camblin et al. showed that IGF-1R, ErbB3, and their ligands are expressed in a significant proportion of ovarian cancer patient samples. Activating the ligands of both IGF-1R and ErbB3 promotes ovarian cancer cell proliferation and pro-survival signaling activation, whereas the dual blocking of IGF-1R and ErbB3 enhances the efficacy of relevant chemotherapies [37]. Another study from Camblin et al. provides evidence for an interplay between IGF-1R and ErbB3 in pancreatic cancer. ErbB3 upregulation may compensate for the IGF-1R blockade and vice versa. They constructed a fully human bispecific tetravalent IGF-1R- and ErbB3-targeting antibody, istiratumab, and found that the addition of istiratumab to the gemcitabine and (nab-) paclitaxel regimen improved chemotherapy activity in vivo [38]. These findings highlight the necessity and effectiveness of the dual inhibition of the EGFR and IGF-1R signaling pathways.

In summary, our study revealed that the EGFR/HER2 inhibitors gefitinib and lapatinib, in combination with the IGF-1R inhibitor linsitinib, have synergistic effects on the inhibition of proliferation, the cell cycle arrest, the apoptosis, the invasion, and the migration of ESCC cells in vitro, as well as on tumor growth in vivo. Our findings support the next clinical exploration of the combination of gefitinib or lapatinib with linsitinib in the treatment of ESCC patients.

## 4. Materials and Methods

### 4.1. Cell Lines and Culture

The human esophageal squamous carcinoma cell (ESCC) lines KYSE150, KYSE450, KYSE510, and TE-7 were obtained from the Cell Center of Peking Union Medical College, China. All cell lines were cultured in an RPMI1640 medium containing 10% FBS, 100 unit/mL penicillin, and 100 µg/mL streptomycin. The cell lines were maintained in a 5% CO_2_ incubator at 37 °C.

### 4.2. Reagents and Antibodies

Lapatinib (GSK572016), gefitinib (ZD1839), and linsitinib (OSI-906) were purchased from TopScience (Shanghai, China), dissolved in dimethyl sulfoxide (DMSO), and stored at a concentration of 10 mmol/L. ((3-(4, 5-Dimethyl-thiazol-2-yl)-2, 5-diphenyltetrazolium bromide) (MTT) was obtained from Sigma-Aldrich Chemical Inc. (St. Louis, MO, USA). The primary antibodies, including phosphorylated -EGFR, -HER2, -IGF-1R, -AKT (Ser473), -p44/42MPAK (ERK), and -β-actin, were provided by Cell Signaling Technology (Danvers, MA, USA). The Annexin V-FITC Apoptosis Detection Kit and cell cycle and apoptosis analysis kit were supplied by Beyotime Biotechnology (Nantong, Jiangsu, China). The Matrigel invasion chamber 24-well plate 8.0 micron was purchased from Corning (Corning, NY, USA).

### 4.3. Cell Viability Assay

Cell viability was measured by MTT assays. ESCC cells were seeded in 96-well plates and cultured for 24 h. After treatment with single drugs (lapatinib, gefitinib, linsitinib) or combinations (lapatinib+linsitinib or gefitinib+linsitinib) at different concentrations for 48 h, 20 µL MTT (5 mg/mL) was added to each well and incubated for 4 h at 37 °C. Formazan was solubilized in 150 µL DMSO, and then the optical density at 570 nm was measured. Growth inhibition was calculated as a percentage of the untreated controls. IC50 values were calculated with the Graphpad Prism Version 9.0 (GraphPad Software Inc, San Diego, CA, USA).

### 4.4. CI Value Calculation

The combination indexes (CI) of lapatinib or gefitinib plus linsitinib were calculated by the median effect principle (Chou–Talalay method). CI < 1, CI = 1, and CI > 1 indicated the synergistic, additive, and antagonistic effect, respectively.

### 4.5. Cell Wound Scratch Assay

ESCC cells were seeded in 6-well plates at a density of 4 × 10^5^ cells per well and cultured for 24 h to yield a confluent monolayer for wounding. A sterile 10 µL pipette tip was used to gently scratch the monolayer at the center of each well. Then, the cells were washed three times with PBS to remove the debris. Lapatinib (2.5 µmol/L), gefitinib (2.5 µmol/L), and linsitinib (10 µmol/L) alone or in combination (lapatinib plus linsitinib or gefitinib plus linsitinib) were added to treat the cells for 24 h and 48 h, and the cells were photographed using a light microscope.

### 4.6. Transwell Migration and Invasion Assay

Cell migration and invasion were measured using a two-chamber transwell system (8 µm pore size). 5 × 10^4^ ESCC cells suspended in 200 µL of serum-free RPMI-1640 medium were planted in the matrigel-coated or uncoated upper chambers of the transwell system. Single lapatinib (2.5 µmol/L), gefitinib (2.5 µmol/L), or linsitinib (10 µmol/L) and the drug combinations were added to the upper chambers. Then, 500 µL of the RPMI-1640 medium containing 10% serum was added to the lower chambers. After being incubated at 37 °C for 48 h, the cells that did not migrate or invade through the filter and/or matrigel were gently wiped with a cotton swab. The migrated or invaded cells beneath the filter were fixed with 4% paraformaldehyde for 30 min and stained with crystal violet for 30 min at room temperature. The chambers were observed under an inverted microscope, and 10 visual fields were randomly selected to count the cells.

### 4.7. Cell Apoptosis Assay

Cell apoptosis was performed with an Annexin V-fluorescein isothiocyanate (FITC)/propidium iodide (PI) staining kit (Beyotime Biotechnology, Jiangsu, China). According to the manufacturer’s instruction, the cells were seeded in 6-well plates and incubated for 24 h, followed by the single drug or the combination treatment for an additional 48 h. The cells were collected, centrifuged, washed twice with PBS, resuspended in 400 µL binding buffer, and then incubated with 5µL Annexin V-FITC and 10 µL PI at room temperature for 15 min in the dark. The cells were analyzed for fluorescence with a flow cytometer (BD Corp).

### 4.8. Cell Cycle Analysis

5 × 10^5^ cells were planted in 60 mm dishes and incubated for 24 h, followed by treatment with lapatinib (2.5 µmol/L), gefitinib (2.5 µmol/L), linsitinib (10 µmol/L), and lapatinib or gefitinib combined with linsitinib for an additional 48 h. The cells were digested with trypsin, washed with PBS, and fixed with cold 70% ethanol; then, they were incubated with staining buffer supplemented with RNase A and propidium iodide. The cell fluorescence was detected by a flow cytometer and was analyzed by ModiFit software (Verity Software House Inc, Topsham ME, USA).

### 4.9. Western Blot Assay

ESCC cells (KYSE150, KYSE450, KYSE510, and TE-7) were plated into 100 mm dishes at a density of 1 ×10^6^ and cultured for 24 h. The medium was discarded, and a serum-free medium containing lapatinib (2.5 µmol/L), gefitinib (2.5 µmol/L), and linsitinib (10 µmol/L) or the drug combinations (lapatinib + linsitinib and gefitinib + linsitinib) was added and incubated for 48 h. After being stimulated with EGF (50 ng/mL), IGF-1 (50 ng/mL), or both for 30 min, the cells were collected and lysed on ice for 30 min. The total proteins extracted from the cells were quantified using the BCA method. Then, 30 µg total proteins were applied on 10% SDS-PAGE and transferred onto polyvinylidene fluoride (PVDF) membranes (Millipore, MA, USA). The membranes were blocked with 5% non-fat milk for 2 h at room temperature and then incubated with primary antibodies overnight at 4 °C (diluted 1:1000 with TBST, Cell Signaling Technology). After washing with TBST buffer three times, the membranes were incubated with secondary HRP-conjugated antibodies for 1 h at room temperature (diluted 1:4000, Cell Signaling Technology). Immobilon Western Chemiluminescent HRP Substrate (Millipore) was added onto the membranes, and the specific bands were captured by the Amersham Imager 600 system (GE Healthcare, Chicago, IL, USA).

For the xenograft tumor tissues, 0.1 g tissues were cut from the liquid nitrogen frozen tumors and were then homogenized and lysed in RIPA buffer (Beyotime Biotechnology, Jiangsu, China) on ice for 30 min. The remaining steps were the same as those mentioned above.

### 4.10. In Vivo Efficacy Assay

Female BALB/c nude mice (6–8 weeks) were purchased from Vital River Laboratory Animal Technology Co., Ltd. (Beijing, China) and maintained under specific pathogen-free (SPF) conditions. KYSE450 cells (5 × 10^6^) suspended in 200 µL PBS were subcutaneously injected into the right armpit of the nude mice. When the average tumor volume reached 100 mm^3^ (about 10 days later), the mice were randomly divided into six groups (*n* = 6): the control group, lapatinib treatment group, gefitinib treatment group, linsitinib treatment group, lapatinib plus linsitinib treatment group, and gefitinib plus linsitinib treatment group. Lapatinib and gefitinib were dissolved in 0.5% carboxymethyl cellulose/0.1% Tween-80, and linsitinib was dissolved in 25 mmol/L tartaric acid solution. Lapatinib, gefitinib, and linsitinib (200 µL) were administered to the mice by oral gavage at doses of 100 mg/kg, 50 mg/kg, and 30 mg/kg, respectively, six times a week. Mice in the control group received 200 µL vehicle (25 mmol/L tartaric acid and 0.5% carboxymethyl cellulose/0.1% Tween-80) by oral gavage until the average tumor volume exceeded 1000 mm^3^. The tumor size and animal body weight were measured every 3 days, and the tumor volume was calculated by (length × width)^2^/2.

### 4.11. Statistical Analysis

All experiments were repeated at least three times independently. The results are presented as the mean ± SD, and the data were analyzed using the GraphPad Prism 9.0 software. Statistical significance was evaluated using one-way analysis of variance (ANOVA, Tukey’s multiple comparison test) or two-way ANOVA (Bonferroni post-tests). *p* < 0.05 was considered statistically significant.

## Figures and Tables

**Figure 1 ijms-23-10382-f001:**
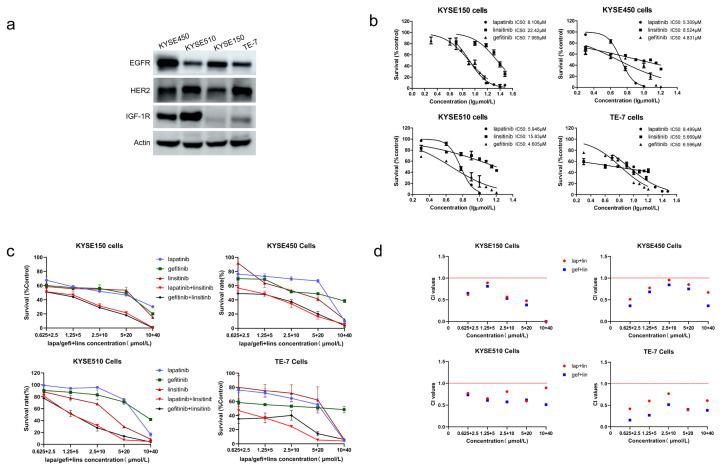
The proliferation inhibition effects of lapatinib, gefitinib, and linsitinib alone or in combination. (**a**) EGFR, HER2, and IGF-1R expression levels on four ESCC cells analyzed by western blot. (**b**) Effects of lapatinib, gefitinib, and linsitinib alone on the proliferation of four ESCC cells measured by MTT assays. (**c**) Effects of lapatinib or gefitinib in combination with linsitinib at the indicated concentrations on the proliferation of four ESCC cells measured by MTT assays. (**d**) The CI values of lapatinib or gefitinib in combination with linsitinib, calculated by using the Chou–Talalay method. All the results were from three independent experiments and are expressed as the mean ± SD.

**Figure 2 ijms-23-10382-f002:**
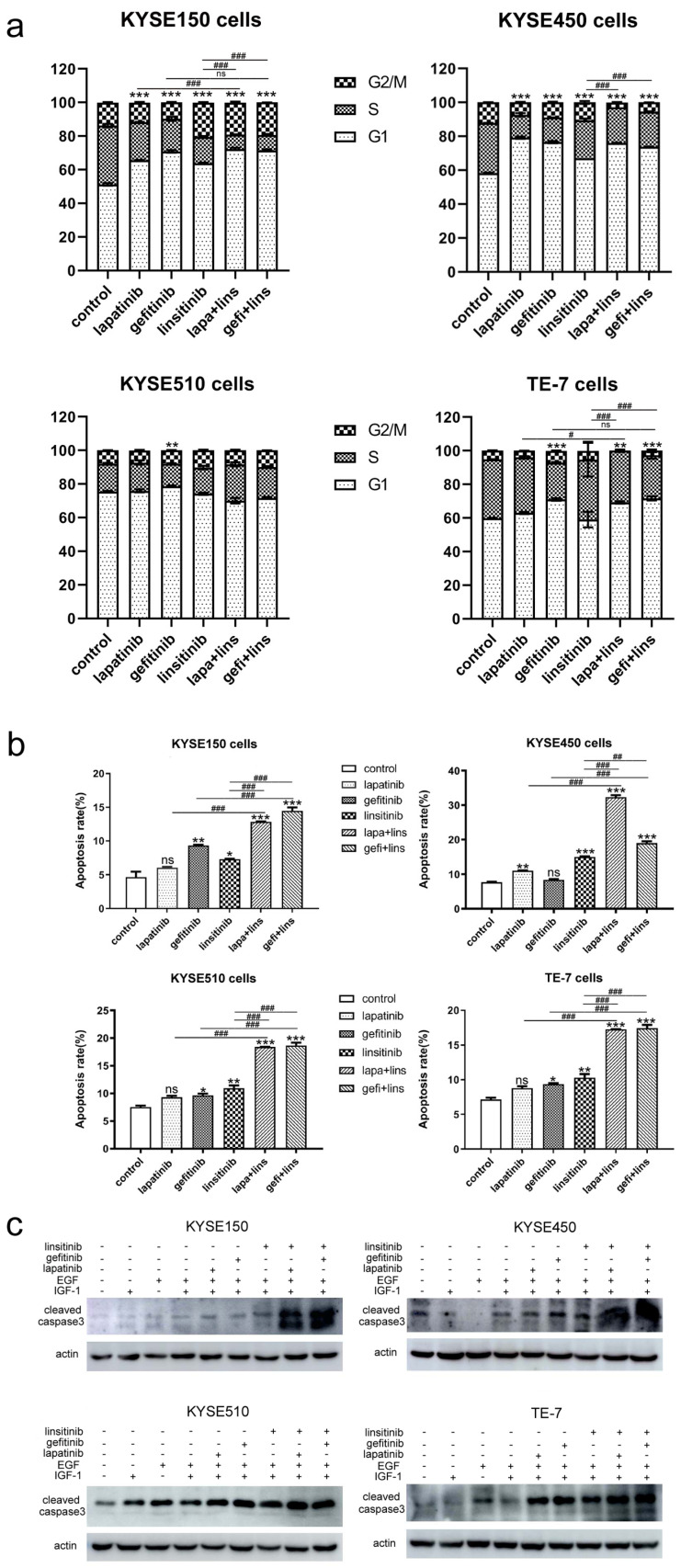
The effects of lapatinib, gefitinib, and linsitinib alone or in combination on the cell cycle distribution and cell apoptosis. (**a**) Four ESCC cells were stained with propidium iodide (PI) after treatment by lapatinib, gefitinib, linsitinib, lapatinib+linsitinib, and gefitinib+linsitinib for 48 h. The fluorescence intensity was measured by a flow cytometer, and the cells distributed in the G1, S, and G2/M phases were analyzed by ModiFit software. ** *p* < 0.01, *** *p* < 0.001, the ratio of G1 phase cells in the drug treatment group vs. the control group. ^#^
*p* < 0.05, ^###^
*p* < 0.001, comparison of the G1 phase ratios between the depicted groups, respectively. (**b**) ESCC cells were exposed to single drugs or in combinations, as indicated, and then the apoptotic cells were stained with Annexin V-FITC and PI. The fluorescence intensity was measured by a flow cytometer. ns, not significant, * *p* < 0.05, ** *p* < 0.01, *** *p* < 0.001 vs. control. ^##^
*p* < 0.01, ^###^
*p* < 0.001, between the depicted groups, respectively. (**c**) Western blot analysis of the expression of cleaved caspase-3 in ESCC cells after treatment with lapatinib, gefitinib, and linsitinib alone or in combination. Actin was used as the loading control.

**Figure 3 ijms-23-10382-f003:**
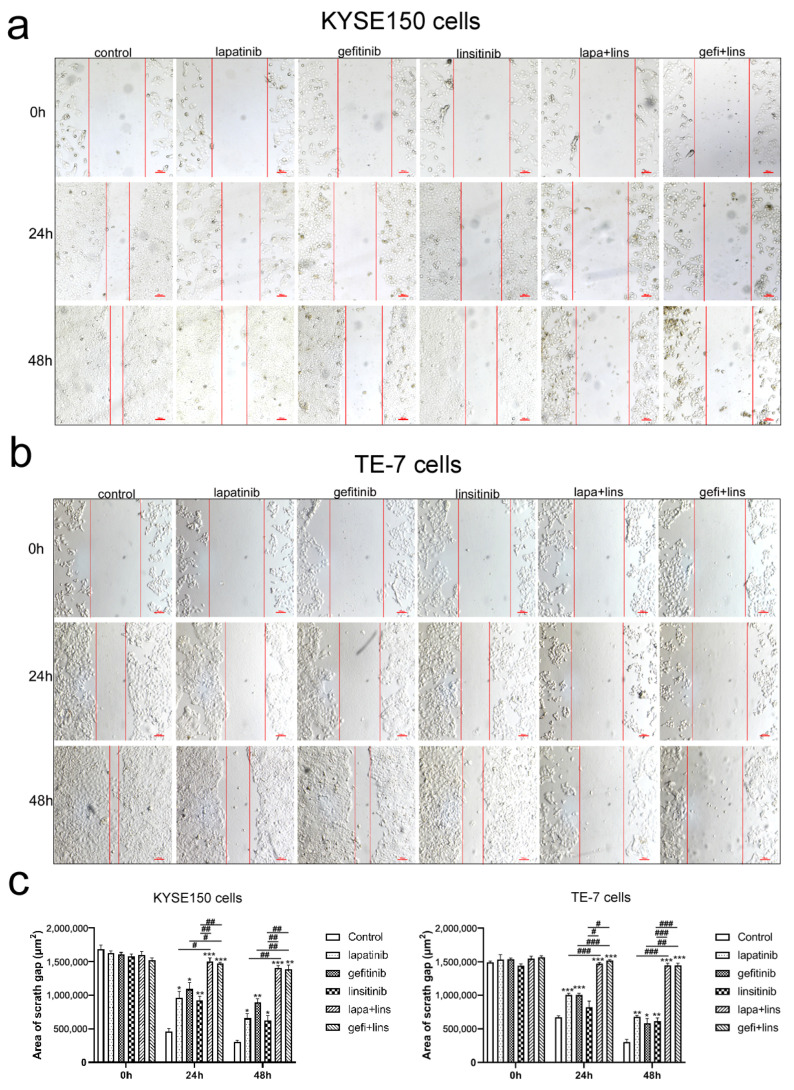
The effects of lapatinib, gefitinib, and linsitinib alone or in combination on the migration of KYSE150 and TE-7 cells detected by the wound healing assay. (**a**,**b**) Representative images of KYSE150 and TE-7 cells. (**c**) Quantification of the area of the scratch of KYSE150 and TE-7 cells by ImageJ. Margins of the scratch gap are indicated with red lines. Scale bars, 100 μm. Statistical significance was calculated based on three independent experiments. * *p* < 0.05, ** *p* < 0.01, *** *p* < 0.001 vs. control. ^#^
*p* < 0.05, ^##^
*p* < 0.01, ^###^
*p* < 0.001 between depicted groups.

**Figure 4 ijms-23-10382-f004:**
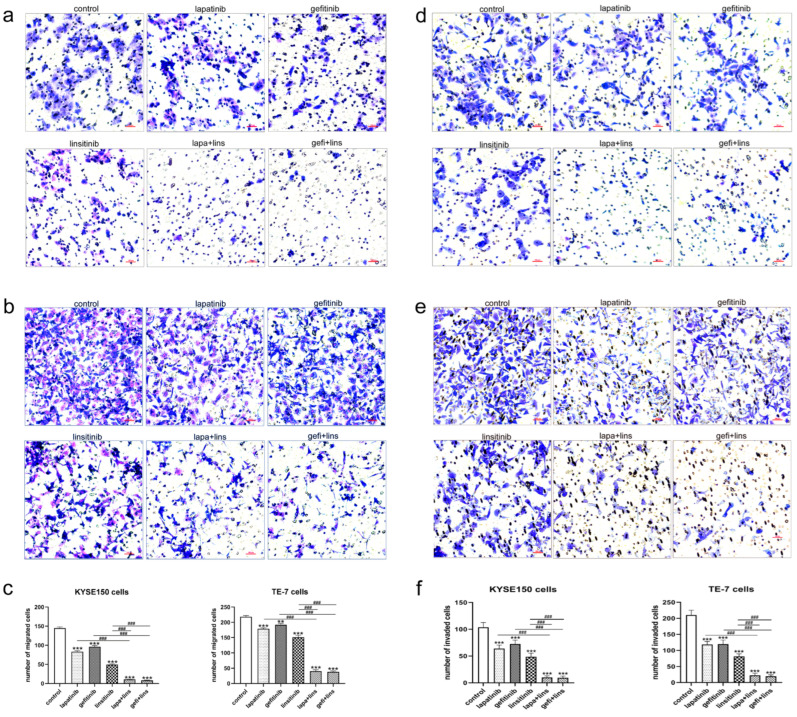
The effects of lapatinib, gefitinib, and linsitinib alone or in combination on the migration and invasion of KYSE150 and TE-7 cells detected by a transwell assay. (**a**,**b**) Representative images of KYSE150 (**a**) and TE-7 (**b**) cells that migrated from the upper chamber membrane of the transwell system. (**c**) The migrated cells from 10 random fields of view (at 200×) were counted. (**d**,**e**) Representative images of KYSE150 (**d**) and TE-7 (**e**) cells that invaded and migrated from the matrigel-coated upper chamber membrane of the transwell system. (**f**) The invaded and migrated cells from 10 random fields of view (at 200×) were counted. Scale bars, 100 μm. ** *p* < 0.01, *** *p* < 0.001 vs. control. ^###^
*p* < 0.001 between depicted groups.

**Figure 5 ijms-23-10382-f005:**
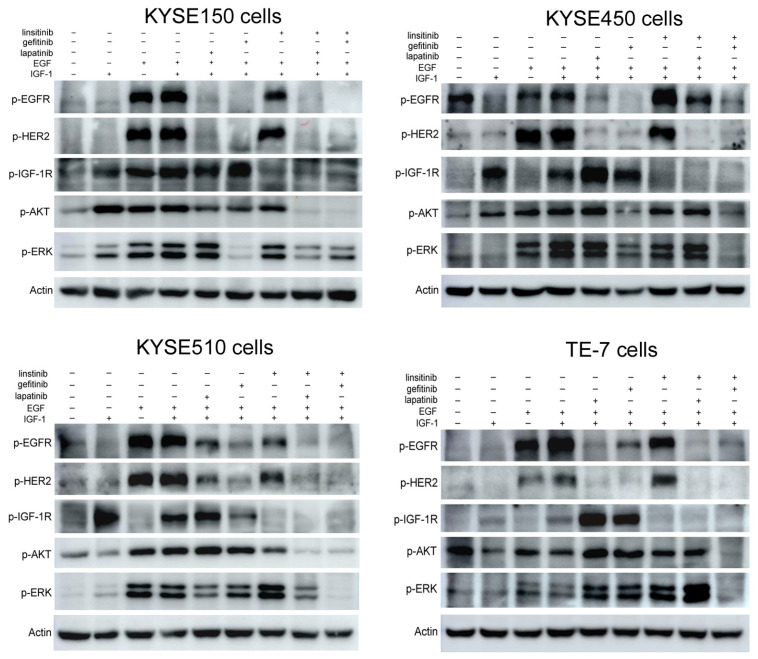
The effects of lapatinib, gefitinib, and linsitinib alone or in combination on the EGFR/HER2 and IGF-1R signaling. Four ESCC cells were serum-starved, treated with single or combined drugs, and stimulated with IGF-1, EGF, or both for 30 min. The phosphorylation of EGFR, HER2, IGF-1R, AKT, and p42/44MAPK (ERK) levels was detected by western blot analysis. β-actin was used as the loading control.

**Figure 6 ijms-23-10382-f006:**
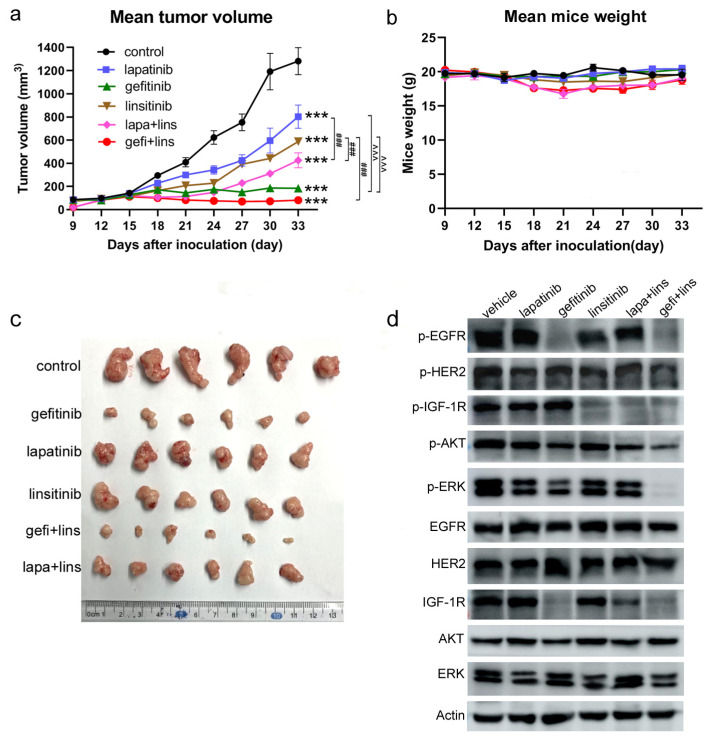
In vivo efficacy of lapatinib, gefitinib, and linsitinib alone or in combination in a KYSE450 xenograft mouse model. KYSE450 cells were subcutaneously inoculated into the right armpit of the nude mice (day 0), and the vehicle, lapatinib (100 mg/kg), gefitinib (50 mg/kg), linsitinib (30 mg/kg), lapatinib plus linsitinib, or gefitinib plus linsitinib treatments were given to the tumor-bearing mice six times a week by oral gavage. The mean tumor volume (**a**) and mean mice weight (**b**) are shown. (**c**) On day 33, the mice were sacrificed, and the tumors were taken from the mice. *** *p* < 0.001 vs. control. ^###^
*p* < 0.001 lapatinib vs. lapatinib+linsitinib, gefitinib vs. gefitinb+linsitinib, linsitinib vs. lapatinib+linsitinib, and linsitinib vs. gefitinb+linsitinib. ^^^^^
*p* < 0.001 gefitinib vs. lapatinib, gefitinib vs. linsitinib. (**d**) The tumor tissues from xenografts were homogenized and lysed, and western blot analysis was performed to detect the phospho-EGFR, -HER2, -IGF-1R, -AKT, and -ERK as well as their total protein levels.

**Table 1 ijms-23-10382-t001:** The IC50 values of lapatinib, gefitinib, or linsitinib alone against four ESCC cells.

Cell Lines	Lapatinib (μmol/L)	Gefitinib (μmol/L)	Linsitinib (μmol/L)
KYSE150	8.106	7.969	22.42
KYSE450	5.309	4.831	8.524
KYSE510	5.946	4.605	15.83
TE-7	8.489	6.596	5.669

**Table 2 ijms-23-10382-t002:** Combination index values of gefitinib or lapatinib in combination with linsitinib in four ESCC cells.

Cell Lines	Combination Index		
	ED50	ED75	ED90
lapatinib			
KYSE150	0.93	0.27	0.08
KYSE450	0.68	0.74	0.81
KYSE510	0.72	0.73	0.75
TE-7	0.51	0.53	0.55
gefitinib			
KYSE150	0.99	0.28	0.08
KYSE450	0.59	0.50	0.55
KYSE510	0.63	0.58	0.55
TE-7	0.29	0.30	0.46

## Data Availability

All data generated or analyzed during this study are included in this published article.

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
