# Peer review of "Dual Inhibition of EGFR and IGF-1R Signaling Leads to Enhanced Antitumor Efficacy against Esophageal Squamous Cancer"

_ijms, 2022, doi:10.3390/ijms231810382_

Round 1

Reviewer 1 Report

In this paper the authors shows that combination of EGFR inhibitor and  IGF-1R inhibitor could strength the inhibition on the activation of EGFR, HER2 and IGF-1R as well as the downstream signaling molecules both in vitro and in vivo, this give us some promising combination strategy for the treatment of ESCC patients.

This manuscript must address the following concerns before the publication.

In figure 1b, label the IC50 of different compounds. and why even with the low expression of IGF-1R(fig1a), the linsitinib has the killing effect in fig1b, and the IC50 is the lowest in TE-7(table 1)?

For Fig2a, it's very messy with the description of this part. I see a little bit of G1 arrest after the combination in KYSE150 but no enhancement in the other 3 cell lines.

In fig2b you stained the cells with the Annexin V-FITC and PI, what is the apoptosis rate representing? Provide the facs figure to show more details.

You mentioned the combination could decrease the migration ability of the cancer cells, but we see there are a lot of cell deaths in fig3 a/b, is it due to the cell death or migration effect that lead to the scratches distance?

In vivo study, provide some other evidence that could show the inhibition. For example, the Pakt/pERK expression by western blot and the Ki67/cleaved caspase3 by the IHC.

For the invasion assay 48 hours would be too long time for the invasion/migration, most of the cells will pass through the well.

In figure5 for the combination group, it seems like gef and lap could decrease the expression of pEGFR but in the combination group it will increase the expression of pEGFR in KYSE450 cells, it is also very messy in this figure the all the cell results are different, and please provide the background protein ERK/EGFR/AKT expression.

In figure 4, the control group, the cells stain is more in d compared to a, but the calculation is more in c than f.

Reviewer 2 Report

This manuscript describes the contents of an in vitro and in vivo study on the anticancer effect of inhibition of EGFR & IGF-1R signaling on esophageal squamous cell carcinoma. I have some comments for the author. ã€€

Comment 1. Is "toxic sings" on page9, line 254 a mistake of "toxic signs"?

Comment 2. Some results seem inconsistent. For example, in Figure 2a, the effect of G1 arrest on KYSE450 cells does not seem to differ between the lapa+lins group and the gefi+lins group, but in Figure 2b, the apoptosis-inducing effect is greater in the lapa+lins group than in the gefi+lins group. appears to be significantly higher than What is your interpretation of this point?

Comment 3. Regarding comment 2, Figure 2a states that the effect of G1 arrest on KYSE450 cells was higher in the lapa+lins group and gefi+lins group than in the EGFR inhibitor alone group including gefitinib. In 2a, the apoptosis-inducing effect does not seem to differ much between the gefi+lins group and the EGFR inhibitor alone group. What is your interpretation of this point?

Comment 4. Following the above comment, an in vivo study showed the effect of suppressing tumor volume using a KYSE450 cell graft model, and it was shown that the effect of suppressing tumor growth was high in the gefi+lins group. In addition, it has been shown that the gefi alone group also has a high tumor growth inhibitory effect. On the other hand, the lapa+lins group appears to have a weaker tumor growth inhibitory effect. I think these results do not match the results of Fig.2. What is your interpretation of this point?

Comment 5. Regarding comments 2 to 4, the result of KYSE450 cell was given as an example, but the discrepancy of the result of Fig2a and Fig2b is also seen in other cell lines, so I think that proper consideration is necessary.

Comment 6. The inhibitory effect on invasion and migration has been shown in KYSE150 and TE-7 cells. On the other hand, an in vivo study was performed using KYSE450 cells. Why did other cell lines (i.e. KYSE450 and KYSE510 cells) not show inhibitory effects on invasion and migration? Why didn't in vivo studies similarly show other cell lines?

Comment 7. As for consideration, I felt that the description, such as comparison with other reports based on the results of this experiment, was insufficient.

Round 2

Reviewer 1 Report

Thanks for the author's response and addressing my concerns.

Last question for this manuscript, you explored the combination effect of these two signaling pathway inhibitions, and found that they could affect the cell cycle/ cell death/migration. You did see the related signaling pathway proteins alteration ,but I don't see the direct molecular level data.For example, the dual inhibition could cause more cell death markers cleaved caspase3/parp increase, and similarly with the cell cycle/migration protein markers change.This will be more convincible and the mechnism are more direct and clear.

Reviewer 2 Report

Thank you for answering my comments.

Author Response

Thank you for your suggestion and hard work!